# van der Waals driven anharmonic melting of the 3D charge density wave in VSe$_2$

Josu Diego[1], A. H. Said[2], S. K. Mahatha[3], Raffaello Bianco[1], Lorenzo Monacelli[4,5], Matteo Calandra[5,6,7], Francesco Mauri [4,5], K. Rossnagel [3,8], Ion Errea [1,9,10 ✉] & S. Blanco-Canosa [10,11 ✉]

Understanding of charge-density wave (CDW) phases is a main challenge in condensed matter due to their presence in high-$T$c superconductors or transition metal dichalcogenides (TMDs). Among TMDs, the origin of the CDW in VSe$_2$ remains highly debated. Here, by means of inelastic x-ray scattering and first-principles calculations, we show that the CDW transition is driven by the collapse at 110 K of an acoustic mode at $\mathbf{q}_{CDW} = (2.25\ 0\ 0.7)$ r.l.u. The softening starts below 225 K and expands over a wide region of the Brillouin zone, identifying the electron-phonon interaction as the driving force of the CDW. This is supported by our calculations that determine a large momentum-dependence of the electron-phonon matrix-elements that peak at the CDW wave vector. Our first-principles anharmonic calculations reproduce the temperature dependence of the soft mode and the T$_{CDW}$ onset only when considering the out-of-plane van der Waals interactions, which reveal crucial for the melting of the CDW phase.

[1] Centro de Física de Materiales (CSIC-UPV/EHU), 20018 San Sebastián, Spain. [2] Advanced Photon Source, Argonne National Laboratory, Lemont, IL 60439, USA. [3] Ruprecht Haensel Laboratory, Deutsches Elektronen-Synchrotron DESY, 22607 Hamburg, Germany. [4] Dipartimento di Fisica, Università di Roma La Sapienza, Roma, Italy. [5] Graphene Labs, Fondazione Instituto Italiano di Tecnologia, Genoa, Italy. [6] Dipartimento di Fisica, Università di Trento, Via Sommarive 14, 38123 Povo, Italy. [7] Sorbonne Universités, CNRS, Institut des Nanosciences de Paris, UMR7588, F-75252 Paris, France. [8] Institut für Experimentelle und Angewandte Physik, Christian-Albrechts-Universität zu Kiel, 24098 Kiel, Germany. [9] Fisika Aplikatua 1 Saila, Gipuzkoako Ingeniaritza Eskola, University of the Basque Country (UPV/EHU), San Sebastián, Spain. [10] Donostia International Physics Center (DIPC), 20018 San Sebastián, Spain. [11] IKERBASQUE, Basque Foundation for Science, 48013 Bilbao, Spain. ✉email: ion.errea@ehu.eus; sblanco@dipc.org

The study of electronic ordering and charge-density-wave (CDW) formation is attracting massive efforts in condensed matter physics[1]. In particular, its dynamical nature is the focus of a strong debate in correlated oxides and high-$T_c$ superconducting cuprates[2], where fluctuations of the charge order parameter[3], dispersive CDW excitations[4], and phonon anomalies[5] are observed. Microscopically, the subtle balance between electron–phonon interaction (EPI) and nested portions of the Fermi surface (singularities in the electronic dielectric function, $\chi_q$, at $\mathbf{q}_{CDW} = 2k_F$) determines the origin and stabilization of the charge periodicities[6]. While the Fermi surface nesting scenario survives for 1D and quasi-1D systems (Peierls transition), its role in higher dimensions remains largely questioned[7,8].

Among the solids showing electronic charge ordering, layered transition metal dichalcogenides (TMDs) represent the first crystalline structures where 3D CDWs were discovered[9]. 1$T$-VSe$_2$ (space group $P\bar{3}$m1) belongs to the series of layered TMDs that develops a 3D-CDW as a function of temperature, $T_{CDW} = 110$ K. However, unlike the isostructural 1$T$-TiSe$_2$, which adopts a commensurate $2 \times 2 \times 2$ CDW ordering with $\mathbf{q}_{CDW} = (0.5\ 0\ 0.5)$ r.l.u[10], 1$T$-VSe$_2$ develops a more complex temperature dependence 3D incommensurate pattern in its CDW phase with a $\mathbf{q}_{CDW} = (0.25\ 0\ -0.3)$ r.l.u CDW wave vector[11], modulating the interlayer distances. 1$T$-VSe$_2$ is rather unique among the 1$T$-polytypes because it develops anomalies in its transport properties and magnetic susceptibility[12] that more closely resemble those of 2$H$-polytypes ($T_{CDW}[2H$-NbSe$_2] = 33$ K, $T_{CDW}[2H$-TaSe$_2] = 122$ K) and presents the lowest onset temperature among them, i.e., $T_{CDW}[1T$-TiSe$_2] = 200$ K, $T_{CDW}[1T$-TaS$_2] = 550$ K[11]. The sizable difference between $T_{CDW}[1T$-VSe$_2]$ and its 1$T$ counterparts can be attributed to the occurrence of large fluctuation effects that lower the mean-field transition temperature[13] or to the out-of-plane coupling[14] between neighboring VSe$_2$ layers assisted by the weak short-range van der Waals interactions[15]. Moreover, the theoretical input based on ab initio calculations is also limited for all these TMDs undergoing CDW transitions due to the breakdown of the standard harmonic approximation for phonons, which cannot explain the stability of the high-temperature undistorted phases[16]. This hinders the study of both the origin and the melting of the electronically modulated state, complicating the comprehensive understanding of the CDW formation.

From the electronic point of view, angle-resolved photoemission (ARPES) experiments in VSe$_2$ reported asymmetric dogbone electron pockets centered at $M(L)$[17] that follow the threefold symmetry of the Brillouin zone (BZ) interior, with nesting vectors closely matching those observed by x-ray scattering[18]. The formation of the CDW results from the 3D warping of the Fermi surface in the $ML$ plane (Fig. 1a shows the high-symmetry points of the Brillouin zone of the hexagonal lattice of VSe$_2$). Moreover, photoemission data also find a partial suppression of the density of states near $E_F$ on the nested portion below 180 K, indicating that a pseudogap opens at the Fermi surface[19]. However, a detailed investigation of the electronic structure is complicated by the 3D nature of the CDW order, and the momentum dependence of the EPI and the response of the lattice to the opening of the gap at $E_F$ remains unsolved. In fact, inelastic x-ray scattering (IXS) and theoretical calculations discarded the Fermi surface nesting scenario proposed for 2$H$-NbSe$_2$[20,21] and 1$T$-TiSe$_2$[22,23] and emphasized the critical role of the momentum dependence of the EPI. In addition, it has been recently demonstrated that large anharmonic effects are required to suppress the CDW phases in TMDs and understand their phase diagrams, both in the bulk and in the monolayer limit[16,24–27]. Indeed, an evolution from the $(4 \times 4)$ CDW in bulk VSe$_2$ to a $(\sqrt{7} \times \sqrt{3})$ electronic reconstruction has been reported by means of scanning tunneling microscopy[28], imperatively calling for a comprehensive description of the nature of the 3D CDW in VSe$_2$.

## Results

**Quasi-elastic central peak.** Figure 1b displays the temperature dependence of the elastic signal at the critical wave vector $\mathbf{q}_{CDW} = (2.25\ 0\ 0.7)$ r.l.u upon cooling from 300 K. The elastic line due to incoherent scattering is barely visible at high temperature and is temperature independent down to 150 K, implying low structural disorder. Below 150 K (Supplementary Fig. 4), a smooth increase of the *quasi*-elastic intensity is observed at $\mathbf{q} = \mathbf{q}_{CDW}$ due to low-energy critical fluctuations and displays a sharp onset at the CDW transition $T \approx 110$ K. No indications of charge instabilities were observed along the $\Gamma \rightarrow M$ and $\Gamma \rightarrow L$ directions. The mean-field critical exponent obtained in the disordered phase at $T > T_{CDW}$, $\gamma = 1.303 \pm 0.004$, is consistent with the existence of a 3D regime of critical fluctuations of an order parameter of dimensions $n = 2$, as expected for a classical $XY$ universality class[29]. A similar critical exponent has been observed in the quasi-1D conductor blue bronze K$_{0.3}$MoO$_3$[30] and ZrTe$_3$[31], which develops a giant Kohn anomaly at the CDW transition.

**Experimental and theoretical phonons.** Figure 1c displays the momentum dependence of the inelastic spectra at $(2 + h\ 0\ 0.7)$ r.l.u. for $0.15 < h < 0.45$ at 300 K. Optical phonons appear above 17 meV and do not overlap with the acoustic branches. At all momentum transfers, $0 < h < 0.5$, the spectrum consists of 2 phonons, labeled as $\omega_1$ and $\omega_2$ in Fig. 1d, in good agreement with the results of the theoretical calculations (see Supplementary Fig. 5 for a precise description and assignment of the 2 branches). The third acoustic mode is silent in IXS as its polarization vector is perpendicular to the wave vector. Both $\omega_1$ and $\omega_2$ belong to the same irreducible representation and, thus, do not cross. For $h < 0.2$, $\omega_1$ develops more spectral weight than $\omega_2$ and, for $h > 0.2$, the intensity of $\omega_2$ increases and $\omega_1$ leads an apparent asymmetric broadening of $\omega_2$, as depicted in Fig. 1d. To obtain quantitative information of the frequency and the phonon lifetime, the experimental scans were fitted using standard damped harmonic oscillator functions convoluted with the experimental resolution of $\approx 1.5$ meV (see Fig. 1d and Supplementary Fig. 6 for a detailed analysis of the fitting). The frequencies of the low-energy acoustic branches $\omega_1$ and $\omega_2$ start around 4 and 8 meV, respectively, and end at $\approx 13$ meV. Remarkably, the results of our ab initio anharmonic phonon calculations with the stochastic self-consistent harmonic approximation (SSCHA)[32–34], which are performed with forces calculated within density-funcitonal theory (DFT) and including van der Waals interactions, show that both $\omega_1$ and $\omega_2$ do not follow a sinusoidal dispersion, but develop a dip at $h \approx 0.25$ r.l.u. The theoretical dispersion nicely matches the experimental data from the zone center to the border of the Brillouin zone (BZ), as shown in Fig. 1e. In fact, the results of the harmonic phonon calculations indicate that the high-temperature structure of 1$T$-VSe$_2$ is unstable towards a CDW transition. It is clear, thus, that anharmonicity stabilizes 1$T$-VSe$_2$ at high temperatures. On the other hand, the linewidth extracted from the analysis (Fig. 1f, symbols) of the $\omega_2$ mode is resolution limited across the whole BZ. Nevertheless, the linewidth of the $\omega_1$ branch is no longer resolution limited between $0.2 < h < 0.3$ r.l.u. and develops an anomalously large broadening of $\approx 4$ meV at $h = 0.25$ r.l.u. Again, the experimental broadening is well captured by our calculations (dashed lines in Fig. 1f), indicating that the large enhancement of the broadening is mainly due to the EPI even if the anharmonic contribution to the linewidth also peaks at $h = 0.25$ r.l.u. (Supplementary Fig. 10).

Given the observation of the phonon broadening at room temperature and the good agreement between theory and experiment, we proceed with the analysis of the lattice dynamics at lower temperatures. At 250 K, the phonon with energy $\approx 7$ meV ($\omega_2$)

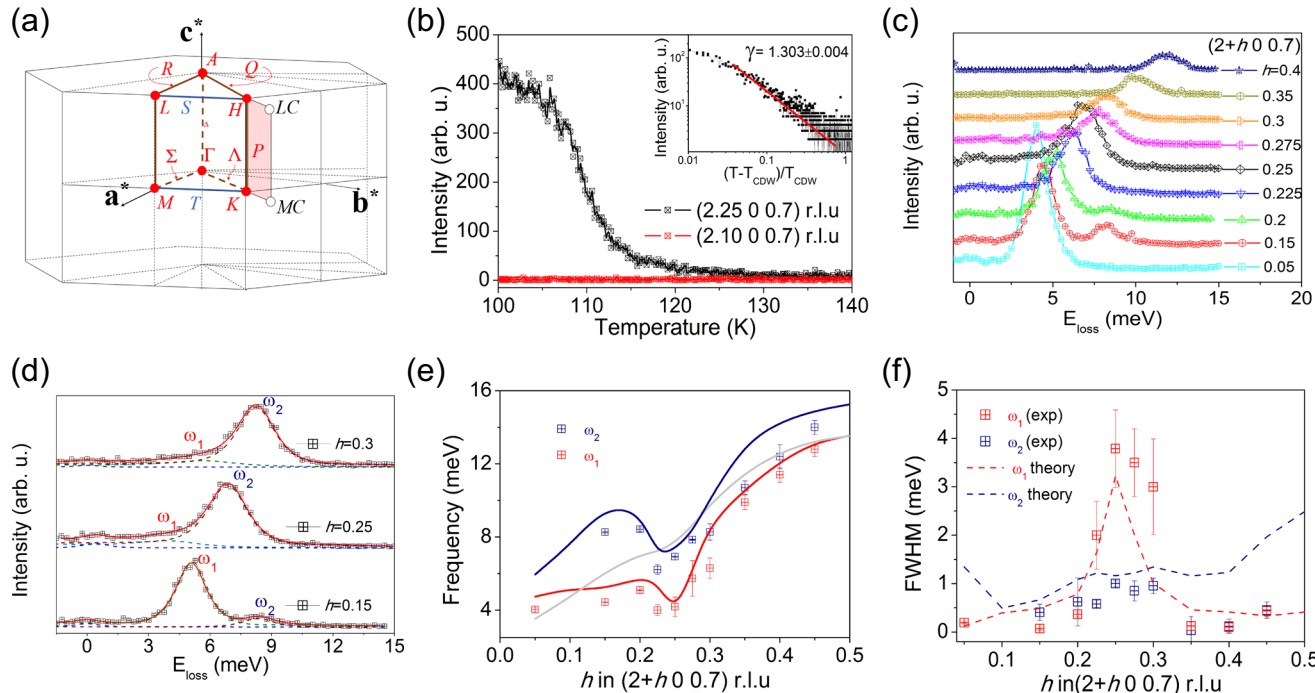

**Fig. 1 Elastic peak, CDW, and lattice dynamics at 300 K. a** The Brillouin zone of space group $P\bar{3}m1$ (164)[52]. **b** Temperature dependence of the elastic line at (2.25 0 0.7) r.l.u. (reciprocal lattice unit) showing the onset of the CDW at 110 K. Inset, scaling analysis of the elastic peak. **c** Energy-momentum dependence of the low-energy acoustic phonons at 300 K from 0.05 < h < 0.4 r.l.u. along the (2 + h 0 0.7) direction. The spectra are vertically offset for clarity. **d** Close-up view of the phonon fitting at 300 K for selected momentum transfers, identifying the $\omega_1$ and $\omega_2$ phonons. **e** Experimental (points) and calculated (solid lines) dispersion of the low-energy acoustic phonons at 300 K. The gray line stands for the silent mode, not observed experimentally. **f** Momentum dependence of the linewidth for $\omega_1$ and $\omega_2$ obtained from the fitting of the experimental spectra to damped harmonic oscillators. In panels (**e**) and (**f**), the error bars represent the fit uncertainty. The calculated linewidth including the contribution of the electron–phonon interaction and anharmonicity is shown as dashed lines.

shows a clear asymmetric broadening at $\mathbf{q}_{CDW}$, i.e, the corresponding branch $\omega_1$ appears to develop a redshift as a function of temperature (Fig. 2b). The dispersion of $\omega_2$ at 150 K is similar to the one at 300 K. Contrarily, $\omega_1$ lowers its energy, softening from room temperature down to 110 K. The softening extends over a wide region of momentum space $0.225 < h < 0.3$ r.l.u. (0.15 Å$^{-1}$) at 150 K, see green dotted line in Fig. 2a (and Supplementary Fig. 8). The pronounced instability of this acoustic mode and its broad extension in momentum space are consistent with the results of our anharmonic phonon calculations (solid lines in Fig. 2c). The momentum space spread of the softening indicates a substantial localization of the phonon fluctuations in real space due to the EPI, questioning the pure nesting mechanism suggested by ARPES[17]. More importantly, the softening of this branch represents the first indication of the lattice response to the formation of the 3D-CDW in VSe$_2$. The analysis of the linewidth reveals that the lifetime of $\omega_2$ remains nearly constant across the BZ and is resolution limited (Fig. 2d). On the other hand, the softening of the $\omega_1$ mode at 150 K is accompanied by an enhancement of the linewidth, as shown in Fig. 2d (6 meV linewidth at 120 K, Fig. 3f) and, again, well modeled by the ab initio calculations (dashed lines in Fig. 2d).

At the critical temperature, $T_{CDW} = 110$ K and $\mathbf{q} \approx \mathbf{q}_{CDW}$, the spectrum is dominated by an elastic central peak at zero energy loss (FWHM = 0.05 r.l.u. and $\Delta E = 1.6$ meV), thus, the soft mode is no longer resolvable (see Fig. 3a–d and Supplementary Fig. 9). Figure 3e displays the temperature dependence of the soft mode, $\omega_1$, as well as the frequency of the phonon obtained ab initio with and without including van der Waals corrections. As plotted in Fig. 3e, the phonon frequency obtained ab initio follows the temperature dependence of the experimental acoustic branch. Moreover, the high temperature 1T structure of VSe$_2$ remains

unstable at all temperatures after withdrawing the van der Waals functional from DFT calculation (see blue triangle in Fig. 3e). The softening of the acoustic phonon is accompanied by a linewidth broadening at $T_{CDW}$ (Fig. 3f).

## Role of EPI

Having achieved a comprehensive description of the CDW and its temperature dependence, we address the crucial role of the EPI and nesting mechanism in the formation of the charge modulated state. In Fig. 4, we plot the calculated harmonic phonon frequency together with the electron–phonon linewidth of the three acoustic modes along $\mathbf{q} = (h\ 0\ -1/3)$ r.l.u calculated within density-functional perturbation theory (DFPT). As it can be seen, the harmonic phonon instability of $\omega_1$ coincides with a huge increase of its linewidth associated with the EPI. The softening and the increase of the electron–phonon linewidth specially affect the $\omega_1$ mode, which suggests that the electron–phonon matrix elements are strongly mode and momentum dependent and have a strong impact on the real part of the phonon self-energy, which determines the harmonic phonon frequencies[8,21]. This behavior is similar to the one reported for 1T-TiSe$_2$ and 2H-NbSe$_2$[20,22]. The real part of the non-interacting susceptibility $\chi_0(\mathbf{q})$, which captures the full Fermi surface topology and also affects the real part of the phonon self-energy (see Supplementary Information), has a softening of around 4% at $\mathbf{q}_{CDW}$, which seems insufficient to explain the large softening of the $\omega_1$ mode. This suggests that the electron–phonon matrix elements are crucial to induce the harmonic softening and that the topology of the Fermi surface is not the driving mechanism. In order to further clarify the point, we calculate the so-called nesting function $\zeta(\mathbf{q})$ which measures the topology of the Fermi

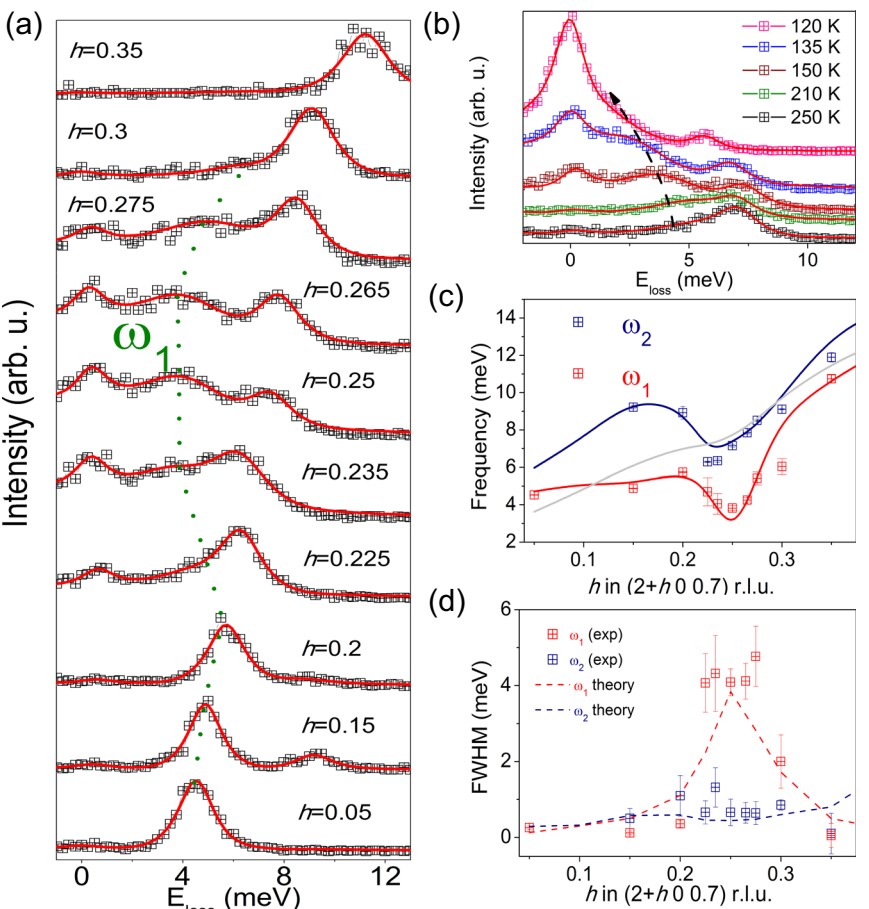

**Fig. 2 Lattice dynamics at 150 K. a** IXS energy-loss scans at $(2 + h\ 0\ 0.7)$ r.l.u for $0.15 < h < 0.35$ and 150 K. The dotted green line follows the dispersion of the soft phonon mode, $\omega_1$ (see text). Each spectrum is vertically shifted for clarity. **b** Energy-loss scans as a function of temperature at $(2.25\ 0\ 0.7)$ r.l.u. The black arrow follows the softening of the low-energy acoustic mode upon cooling. In both (**a**) and (**b**) red lines are the result of the fitting to damped harmonic oscillator functions convoluted with the instrumental resolution. **c** Momentum dependence of the frequency of the $\omega_1$ and $\omega_2$ branches at 150 K. The anharmonic phonon dispersions of the acoustic modes obtained at 150 K are plotted as solid lines. The gray line represents the acoustic mode that is silent in IXS. **d** Experimental (symbols) and theoretical (dashed lines) momentum dependence of the linewidth for $\omega_1$ and $\omega_2$. The error bars represent the fit uncertainty. The theoretical calculation accounts for both the electron–phonon and anharmonic contributions to the linewidth.

surface and peaks at the nesting **q** points (Supplementary Information). As shown in Fig. 4c, it peaks at $\mathbf{q}_{CDW}$, which indicates that the CDW vector coincides with a nested region of the Fermi surface. Considering that for constant electron–phonon matrix elements the nesting function coincides with the phonon linewidth given by the EPI, it is illustrative to compare them. Clearly, the phonon linewidth of the $\omega_1$ mode coming from the EPI depends much more drastically on momentum than the nesting function: it changes by orders of magnitude as a function of **q** while the nesting function only by less than a factor of two. This is highlighted in the ratio between the linewidth and the nesting function plotted in Fig. 4d, which measures the momentum dependence of the electron–phonon matrix elements and should be flat if the electron–phonon matrix elements were constant. This ratio depends much more strongly on momentum than the nesting function itself and resembles the linewidth dependence, underlining again that the momentum dependence of the electron–phonon matrix elements plays a crucial role here. In conclusion, the EPI is the main driving force of the CDW transition in 1T-VSe$_2$ despite the presence of nesting at $\mathbf{q}_{CDW}$. Nevertheless, the **q**-range over which the phonon softens, $\Delta\mathbf{q} \approx 0.075$ r.l.u., even if it coincides with an increase of the electron–phonon linewidth, is a factor of 3 less than in 1T-TiSe$_2$[22], where EPI and excitonic correlations are responsible for the structural instability and the CDW order,

pointing to an intricate relationship between EPI and Fermi surface nesting scenarios in VSe$_2$.

## Discussion

Our anharmonic calculations, which predict that the $\omega_1$ frequency vanishes between 75 and 110 K, are in good agreement with the experimentally measured phonon frequencies and the CDW temperature onset, $T_{CDW} = 110$ K. When the SSCHA anharmonic calculation is repeated without including the van der Waals corrections (blue triangles in Fig. 3e), the softest acoustic mode at $\mathbf{q}_{CDW}$ remains unstable even at room temperature. Remarkably, the weak van der Waals forces (of the order of ~1mRy/$a_0$ for a typical SSCHA supercell calculation) are responsible for the stabilization of the 1T structure of VSe$_2$ and play a crucial role in melting the CDW. On the other hand, the damping ratio, $\Gamma/\tilde{\omega}_q$, increases upon cooling and the phonon becomes critically overdamped at $\mathbf{q}_{CDW}$ and 110 K. The damping ratio $\Gamma/\tilde{\omega}_q$ is given by $\omega_0 = (\tilde{\omega}_q^2 - \Gamma^2)^{1/2}$, where $\Gamma$ is the linewidth $\tilde{\omega}_q$ is the phonon energy renormalized by the real part of the susceptibility and $\omega_0$ is the energy of the phonon fitted to damped harmonic oscillator. The critical exponent derived from the fitting of the phonon frequency *vs* reduced temperature $((T-T_{CDW})/T_{CDW})$, $\beta = 0.52 \pm 0.04$, agrees with the square-root power law

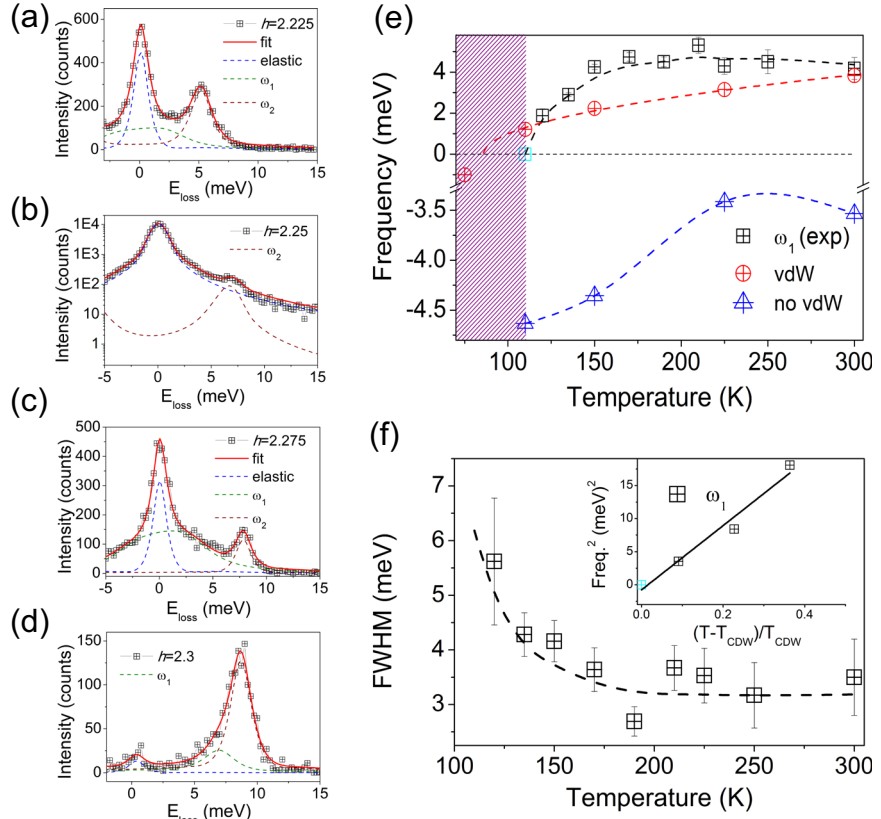

**Fig. 3 Phonon collapse and van der Waals melting. a–d** Representative IXS spectra at 110 K and their corresponding fitting. The IXS scan at $h$=2.25 r.l.u. is presented in logarithmic scale due to the large enhancement of the elastic line. $\omega_1$ stands for the soft mode. **e** Temperature dependence of the energy of the $\omega_1$ branch and the anharmonic theoretical frequencies obtained with and without van der Waals corrections. The shaded area defines the CDW region. **f** Temperature dependence of the linewidth. Inset, squared frequency of the soft mode as a function of the reduced temperature. Lines are guides to the eye. The cyan squares in (**e**) and (**f**) (inset) refer to the frequency of the $\omega_1$ phonon extrapolated to the CDW temperature, since the large enhancement of the elastic line precludes the extraction of its energy from the fitting analysis. The error bars in the experimental data points in panel (**e**) and (**f**) represent the fit uncertainty.

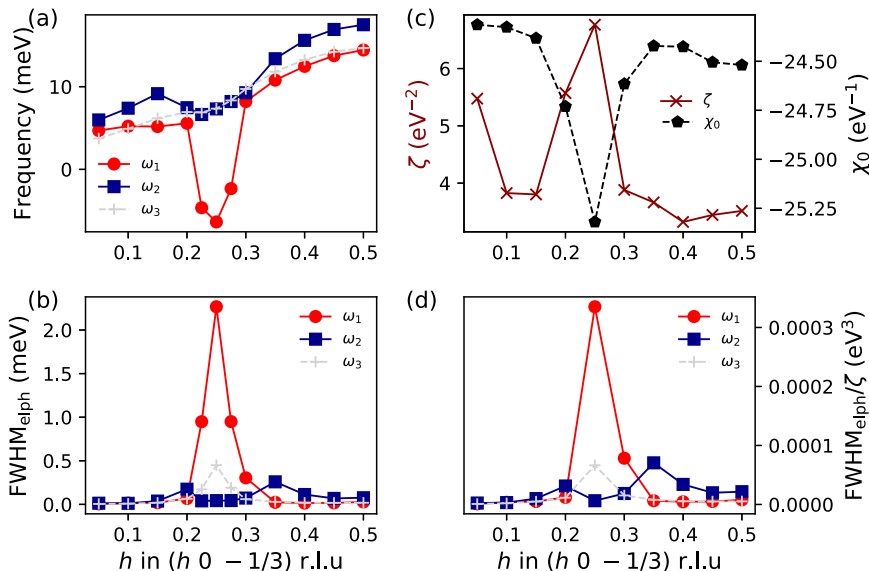

**Fig. 4 Electron–phonon interaction vs Fermi surface nesting. a** Calculated harmonic phonon spectra of 1T-VSe$_2$ along ($h$ 0 -1/3) r.l.u. Only acoustic modes are shown. The gray line denotes the mode silent in IXS, which is labeled as $\omega_3$ here. **b** Phonon linewidth (full width at half maximum) given by the electron–phonon interaction for the same modes. **c** Real part of the non-interacting susceptibility, $\chi_0$, as well as the nesting function, $\zeta$, at the same wave vectors. **d** Ratio between the full width at half maximum given by the electron–phonon interaction and the nesting function.

expected from the mean-field theory (inset of Fig. 3f) and, therefore, fluctuation corrections are unnecessary to invoke the low $T_{CDW}$ of VSe$_2$ as compared to its $1T$ counterparts. The critical role of the EPI has been recently suggested by Raman scattering[35] and DFT calculations[36]. Indeed, revisited ARPES experiments[37] in NbSe$_2$ revealed a pronounced dispersion along $k_z$ discarding the nesting-driven CDW formation and leaving EPI as the major contributor[20]. Although our results indicate an EPI-driven CDW instability, nesting is present and, thus, the charge modulated ground state of VSe$_2$ has to be understood as an interplay between EPI and Fermi surface nesting scenarios.

In conclusion, we have observed with high-resolution IXS that the CDW transition in $1T$-VSe$_2$ is driven by the collapse of an acoustic mode at $\mathbf{q}_{CDW} = (0.25\ 0\ -0.3)$ at $T_{CDW} = 110$ K. The high-temperature $1T$-VSe$_2$ phase is stable thanks to anharmonic effects. The observed wide softening in momentum space, the calculated strongly momentum dependent electron–phonon line-width that peaks at $\mathbf{q}_{CDW}$, and the weaker dependence on the wave vector of the susceptibility suggest that the EPI is the main driving force of the CDW transition despite the presence of nesting. Moreover, the results show that van der Waals forces are responsible for the melting of the CDW. The dominant role of van der Waals forces here may be attributed to the out-of-plane nature of the CDW, which modulates the interlayer distance. This is not the case in $2H$-NbSe$_2$, where the bulk and monolayer transition temperatures seem to be similar[26,38]. This line of thinking is consistent with the enhancement of the CDW in monolayer VSe$_2$, $T_{CDW} = 220$ K[28], since the out-of-plane van der Waals interactions are absent in this case. The critical role of out-of-plane coupling of layers has also been highlighted in the development of the 3D CDW in high-$T_c$ cuprate superconductors[39–41].

## Methods

**Sample growth and characterization**. High-quality single crystals of VSe$_2$ with dimensions $2 \times 2 \times 0.05$ mm$^3$ were grown by chemical vapor transport (CVT) using iodine as transport agent (see Supplementary Figs. 1, 2 for their structural, magnetic[42] and electronic characterization).

**Inelastic x-ray scattering (IXS) measurements**. The high-resolution IXS experiments were carried out using the HERIX spectrometer at the 30-ID beamline of the Advanced Photon Source (APS), Argonne National Laboratory. The incident beam energy was 23.72 keV and the energy and momentum resolution was 1.5 meV and 0.7 nm$^{-1}$, respectively,[43] The components $(hkl)$ of the scattering vector are expressed in reciprocal lattice units (r.l.u.), $(hkl) = h\mathbf{a}^* + k\mathbf{b}^* + l\mathbf{c}^*$, where $\mathbf{a}^*$, $\mathbf{b}^*$, and $\mathbf{c}^*$ are the reciprocal lattice vectors. The experimental lattice constants of the hexagonal unit cell at room temperature are $a = 3.346$ Å, $c = 6.096$ Å, and $\gamma = 120°$. Here, we focus on the low-energy acoustic phonon branches dispersing along the $(0 < h < 0.5\ 0\ -0.3)$ r.l.u direction in the Brillouin zone near the reciprocal lattice vector $\mathbf{G}_{201}$, thus, in the range $(2 + h\ 0\ -0.3)$ r.l.u with $0 < h < 0.5$.

**First-principles calculations**. The variational SSCHA[32–34] method was used to calculate temperature-dependent phonons fully accounting for non-perturbative anharmonic effects. The variational free energy minimization of the SSCHA was performed by calculating forces on $4 \times 4 \times 3$ supercells (commensurate with $\mathbf{q}_{CDW}$) making use of DFT within the Perdew–Burke–Ernzerhof (PBE)[44] parametrization of the exchange-correlation functional. Van der Waals corrections were included within Grimme's semiempirical approach[45]. Harmonic phonon frequencies and electron–phonon matrix elements were calculated within density-functional per-turbation theory (DFPT)[46]. The force calculations in supercells needed for the SSCHA as well as the DFPT calculations were performed within the QUANTUM ESPRESSO package[47,48] (See Supplementary Information for further details on the calculations, which includes citations to refs. [49–51]).

## Data availability

The data that support the findings of this study are available from the corresponding author upon reasonable request. See author contributions for specific data sets.

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

## Acknowledgements

The authors acknowledge valuable discussions with V. Pardo, A. O. Fumega and M. Hoesch. S.B-C thanks the MINECO of Spain through the project PGC2018-101334-A-C22. F.M. and L.M. acknowledge support by the MIUR PRIN-2017 program, project number 2017Z8TS5B. M.C. acknowledges support from Agence Nationale de la Recherche, Project ACCEPT, Grant No. ANR-19-CE24-0028 and M.C and F.M. the Graphene Flagship Core 3. Calculations were performed at the Joliot Curie-AMD supercomputer under the PRACE project RA4956. This research used resources of the Advanced Photon Source, a U.S. Department of Energy (DOE) Office of Science User Facility, operated for the DOE Office of Science by Argonne National Laboratory under Contract No. DE-AC02-06CH11357. Extraordinary facility operations were supported in part by the DOE Office of Science through the National Virtual Biotechnology Laboratory, a consortium of DOE national laboratories focused on the response to COVID-19, with funding provided by the Coronavirus CARES Act.

## Author contributions

S.B.-C. conceived and managed the project. S.K.M. and K.R. synthesized and S.B.-C. characterized the samples. A.H.S. and S.B.-C. carried out the high-resolution IXS experiments. S.B.-C. analyzed the experimental data. J.D., R.B., L.M., M.C., F.M. and I.E. performed the first principles calculations. S.B.-C. and I.E. wrote the manuscript with input from all co-authors.

## Competing interests

The authors declare no competing interests.
