## [Peer Review File · Nature Communications]

REVIEWER COMMENTS

Reviewer #1 (Remarks to the Author):

Manuscript ID: NCOMMS-20-29482

Review report for “Phonon collapse and van der Waals melting of the 3D CDW of VSe₂”, by Diego et. al. submitted for review purpose in Nature Communications.

The manuscript reports on the role of electron phonon coupling in understanding of the charge density wave (CDW) in bulk single crystalline VSe₂. The authors have used high resolution inelastic x-ray scattering (IXS) and DFT calculations to claim that the CDW is driven by ‘the collapse of an acoustic mode at qCDW (2.25 0 0.7) and melting of CDW, thus the electron-phonon interactions (EPI) are important. The softening of acoustic mode starts below 225 K and expands over a wide region of the Brillouin zone, which is identifying the EPI as the driving force of CDW. Authors claim that susceptibility depends weakly on the wave vector and thus electron phonon interactions are driving force despite of Fermi surface nesting.

Similar IXS studies have been reported for NbSe₂ and TiSe₂ [PRL 107, 107403 (2011) and PRL 107, 266401 (2011), respectively]. In case of VSe₂, the present work is important because there is no clear understanding of origin behind CDW formation. Recently, EPI is demonstrated using Raman spectroscopy to understand the CDW in 1T-VSe₂ [Physical Review Research, 02, 033118 (2020)]. While there are several reports suggesting that nesting alone is insufficient to explain the CDW, but hardly any report claiming EPI for CDW in VSe₂. Thus, the manuscript is interesting and the results are significant. However, there are some minor suggestions and comments on the manuscript which have to be addressed before publication.

1. What is the physical significance of CDW melting?
2. While the last line of the abstract clarifies the outcome, the title seems to be misleading. Authors mentioned that that van der Waals forces are responsible for the melting of the CDW. The phrase “van der Waals melting of 3D CDW of VSe₂” is unclear in title.
3. The onset of CDW transition shown in Figure 1 (b) seems to be ~ 120 K rather than 110 K. Here, the crystal quality can play a very critical role in the CDW transition temperature. The mild intercalation of V in VSe₂ is surprisingly present in crystals, which can affect CDW [Phys. Rev. Res., 02, 033118 (2020)]. It would be good to show the quality of the crystal (XRD) and some evidence of the TCDW by transport measurements, in supporting data.

4. Additionally, how the quality of crystal much will affect the IXS? Author may like to comment or discuss on it.

5. In figure 3(a) and 3(c) authors have used three peak fitting whereas for 3(b) two peak fitting is used to demonstrate the absence of ω_1 at $h = 2.25$ at 110 K. But it seems that ω_1 with weak intensity is present in Figure 3(b). Authors can clarify on this.

6. As a general curiosity on the dispersion of ω_1 with h below CDW transition temperature, if the Kohn anomaly can be realized?

Some minor comments

7. Higher font size for text in Figure 1 (a) should be used for better visibility.

8. Some typo errors have been seen in the manuscript, which can be rectified during revisions.

The manuscript requires revisions and clarifications for publications.

Reviewer #2 (Remarks to the Author):

The manuscript "Phonon collapse ..." by Diego et al presents an in-depth analysis of the CDW formation mechanisms in VSe₂. High-resolution experimental IXS data are supported by first-principles calculations of band structure, phonon spectrum including anharmonicity effects, and electron-phonon interaction. The main novelty of this work is the attribution of the CDW formation to the collapse of an acoustic phonon mode whose momentum coincides with the Fermi surface nesting. Other non-trivial findings include the EPI matrix-element effects, the role of anharmonicity in stabilization of high-temperature phase of VSe₂, etc. This is a wonderful in-depth work. I should however bring up to the author's attention a few points which might be elaborated:

- In connection with Fig. 1(b), it might be helpful to discriminate the elastic line from the elastic central peak in the experimental IXS spectrum;

- On page 3, the first paragraph, the authors drop that the momentum-space spread of the phonon mode indicates a substantial localization of the phonon fluctuations. How can such broadening be

separated from that caused, in view of the acoustic-branch dispersion, by a drop of the phonon lifetime due to the EPI outburst near the CDW momentum?

- The authors might be aware of the most recent ARPES work on NbSe₂ [F. Weber et al, Phys. Rev. B 97 (2018) 235122] which identifies clearly 3D character of the Fermi-surface pockets, denying any their nesting;

- My main point of criticism is that the authors try to claim the phonon-mode collapse being the main mechanism of the CDW formation, whereas in reality it is highly entangled with the Fermi-surface nesting. Indeed, the momentum of this phonon itself is found, expectedly, exactly at the nesting vector determined by the electronic subsystem. Furthermore, the nesting by itself has a tendency to increase EPI. Finally, the authors claim that although the nesting-function peak does contribute to the phonon-energy renormalization, the peak of the EPI matrix element gives yet a larger effect; isn't this anomalous behavior of the latter exactly due to the electron-nesting feature of the electronic structure?

While the above points can certainly be elaborated, I am a bit hesitant regarding the publication recommendation. Apart from being too radical on the phonon-collapse role, the work is definitely solid and the results are reliable. However, it is narrowly focused on a particular material VSe₂ without a justified perspective to other cases of general interest. Furthermore, the manuscript is written for specialists and would hardly be accessible for a general reader. I am afraid that in Nature Comm, in the immense of popular biology or genetics papers, this great work would not score as much impact as in a more specialized physics journal such as Comm Phys or Phys Rev Lett.

Reviewer #3 (Remarks to the Author):

The authors report inelastic x-ray scattering measurements of the phonons responsible for the charge density wave (CDW) transition in 1T-VSe₂, along with first-principles calculations to probe the origin of this transition. By studying momentum dependence of the inelastic spectra at different temperatures, one of the two observed acoustic modes is found to soften around the CDW wave vector quickly as the temperature decreases toward T_{CDW} , accompanied by an enhanced linewidth. From the large momentum space spread of the softening and further calculations showing strongly peaked phonon linewidth and electron-phonon matrix elements at the CDW wave vector, the authors conclude that the CDW instability is mainly driven by the electron-phonon interaction as opposed to pure Fermi surface nesting. Moreover, anharmonic phonon calculations suggest that the

weak van der Waals forces are indispensable for describing the stabilization of the high-temperature structure or the melting of 3D CDWs in VSe₂. The present work is of interest for understanding the CDW origin in 1T-VSe₂, but there still remain some questions that need to be addressed, as described below:

(1) On Page 1, the authors refer to the suppression of CDW with temperature in anharmonic phonon calculations as van der Waals melting in the title, based on the calculated result that the TCDW onset can only be reproduced when the van der Waals interactions are treated in the force calculations needed for the anharmonic SSCHA method. However, because the forces in supercells are calculated at $T = 0$ K, the real driving force for the melting of CDWs should still be the phonon-phonon interactions. On the other hand, in DFT-PBE calculations as adopted in this work, the van der Waals correction is essential for studying the properties of layered TMD materials and thus just a necessary ingredient. Furthermore, it is also confusing to attribute the role of van der Waals forces to an out-of-plane nature of the 3D CDW in VSe₂, which is argued to modulate the interlayer distance differently compared to 2D CDW systems like 2H-NbSe₂. To reinforce the so-called van der Waals melting, the current work mentions on Page 5 that anharmonic calculations explain the enhancement of T_{CDW} in monolayer VSe₂ but predict similar T_{CDW} values in bulk and monolayer NbSe₂ (Ref. 27, i.e. arXiv:2004.08147). Since anharmonic data on monolayer VSe₂ are not reported and whether or not the van der Waals correction was already considered in the NbSe₂ work remains unknown, it is hard to understand at present the difference in out-of-plane coupling between 3D and 2D CDW systems and its effect on the CDW transition.

(2) In the calculation details section of the supplementary information, it is noted that a mixed potential is used in DFT, phonon spectrum and electron-phonon calculations, i.e., a ultrasoft pseudopotential for V and a norm-conserving pseudopotential for Se. In consideration of the transferability of the potential and the fact that most theoretical works use pseudopotentials constructed for all atoms in a system within the same framework, the reasons why the authors adopt such a mixed potential need to be explained.

Following the referee's comments, we are re-submitting manuscript including the revisions of the authors, highlighted in blue. In addition, we have added a few comments in the main text, which we summarize here:

- We have modified the title of the paper as suggested by the referee 1.
- Following the advice of Reviewer #2, we have stressed in the introduction section the importance of our work in relation with the hot research field of monolayers of TMDs.
- Figure 3e and 3f (inset) has been modified. The changes are explained in the caption of the figure 3 and in the response to the question 5 of the referee 1.
- The elastic central peak section has been reformulated to better explain the difference between the elastic line and the elastic central peak.

Following referees' advices, we have now discussed 2 papers in the discussion section: Physical Review Research 02, 033118 (2020) and Phys. Rev. B 101, 235405 (2020). The discussion section has been, therefore, updated.

In order to explicitly visualize the changes made to the manuscript, we attach to this letter an explicit comparison of the resubmitted paper with the original one.

Below, we address the referee's questions.

Reviewer #1

Review report for "Phonon collapse and van der Waals melting of the 3D CDW of VSe₂", by Diego et. al. submitted for review purpose in Nature Communications.

The manuscript reports on the role of electron phonon coupling in understanding of the charge density wave (CDW) in bulk single crystalline VSe₂. The authors have used high resolution inelastic x-ray scattering (IXS) and DFT calculations to claim that the CDW is driven by 'the collapse of an acoustic mode at qCDW (2.25 0 0.7) and melting of CDW, thus the electron-phonon interactions (EPI) are important. The softening of acoustic mode starts below 225 K and expands over a wide region of the Brillouin zone, which is identifying the EPI as the driving force of CDW. Authors claim that susceptibility depends weakly on the wave vector and thus electron phonon interactions are driving force despite of Fermi surface nesting.

Similar IXS studies have been reported for NbSe₂ and TiSe₂ [PRL 107, 107403 (2011) and PRL 107, 266401 (2011), respectively]. In case of VSe₂, the present work is important because there is no clear understanding of origin behind CDW formation. Recently, EPI is demonstrated using Raman spectroscopy to understand the CDW in 1T-VSe₂ [Physical Review Research, 02, 033118 (2020)]. While there are several reports suggesting that nesting alone is insufficient to explain the CDW, but hardly any report claiming EPI for CDW in VSe₂. Thus, the manuscript is interesting and the results are significant. However, there are some minor suggestions and comments on the manuscript which have to be addressed before publication.

Response:

We thank the referee for his/her positive review and the critical reading of the manuscript. Moreover, we thank him/her from bringing up a new reference, which we now discuss in the main text. Below, we will give a reply to his/her questions.

1. What is the physical significance of CDW melting?

We simply use the word '*melting*' as a synonymous of phase transition. The sentence 'CDW melting' refers to the disappearance of the charged modulated state towards an electronically and structurally more uniform state, the high-temperature high-symmetry phase. The word '*melting*' is not uncommon to refer to CDW formation/disappearance and has been widely used in the literature. See for instance:

- Charge Density Wave Melting in One-Dimensional Wires with Femtosecond Subgap Excitation. Phys. Rev. Lett. 123, 036405 (2019).
- Charge-density-wave melting in the one-dimensional Holstein model. Phys. Rev. B 101, 035134 (2020).
- Femtosecond x rays link melting of charge-density wave correlations and light-enhanced coherent transport in $\text{YBa}_2\text{Cu}_3\text{O}_{6.6}$ Phys. Rev. B 90, 184514 (2014).
- Spectroscopic fingerprint of charge order melting driven by quantum fluctuations in a cuprate Nature Physics (2020) <https://doi.org/10.1038/s41567-020-0993-7>

2. While the last line of the abstract clarifies the outcome, the title seems to be misleading. Authors mentioned that that van der Waals forces are responsible for the melting of the CDW. The phrase "van der Waals melting of 3D CDW of VSe_2 " is unclear in title.

We understand that the referee finds the title a bit confusing. Another referee also criticized this point. Indeed, the CDW melts due to the entropy contribution of anharmonic phonon-phonon interactions. However, this anharmonic interaction is affected by van der Waals interactions, since unless van der Waals forces are considered, the CDW does not melt below room temperature (see Fig. 3e). Therefore, van der Waals forces are crucial to melt the CDW. In order to not confuse the reader and simplify the message, we have decided to update the title of the paper to "Van der Waals driven anharmonic melting of the 3D charge density wave in VSe_2 ".

3. The onset of CDW transition shown in Figure 1 (b) seems to be ~ 120 K rather than 110 K. Here, the crystal quality can play a very critical role in the CDW transition temperature. The mild intercalation of V in VSe_2 is surprisingly present in crystals, which can affect CDW [Phys. Rev. Res., 02, 033118 (2020)]. It would be good to show the quality of the crystal (XRD) and some evidence of the T_{CDW} by transport measurements, in supporting data.

We thank the referee for his/her comment and agree with him/her. In fact, the authors are fully aware of the effect of the Se vacancies on the CDW of VSe_2 . As reported by Fumega et al. in Journal of Physical Chemistry C 123 (45), 27802-27810 (2019), the CDW is strongly reduced by 65 K with 10% Se deficiency. In order to check the quality of the sample, we show in the Supplementary Information the single crystal x-ray pattern and the electron dispersive x-ray (EDX) analysis. The EDX analysis indicates a V:Se=1:2 ratio (atomic percentage 33.71% of V and 66.29% of Se) and the c-axis lattice parameter obtained from the x-ray diffraction $c = 6.906$ Å, agrees with the one reported in literature. Besides the magnetization measurements presented by Fumega et al. and reproduced in the supplementary information (Fig. S2), additionally, we show the temperature dependence of the resistance in a VSe_2 flake. The T_{CDW} , defined following the criteria reported in Physical Review Research 02, 033118 (2020) is 110-112 K, according to the minimum of the derivative of the resistance.

4. Additionally, how the quality of crystal much will affect the IXS? Author may like to comment or discuss on it.

Inelastic x-ray scattering (IXS) is an ideal technique to check the quality of the crystal. A highly disordered single crystal (defects, Se deficiency...) usually gives an enhancement of the elastic line ($E_{\text{loss}}=0$), which is temperature independent. As can be seen in the supplementary information Figs. S4 and S6, at momentum transfers away from the q_{CDW} , the elastic line is nearly absent, which is a clear indication of the high quality of our crystals. In addition, disorder leads to a loss of phonon coherence and the increase of its linewidth. As reported throughout the manuscript, the ω_2 phonon is always resolution limited ($\Delta E=1.5$ meV), which again indicates a high crystallinity. Below, we present the IXS spectra measure along the $(h\ 0\ 4)$ direction $h=0.25$ r.l.u. at 50 K, showing an extremely narrow (resolution limited) transversal acoustic phonon with a linewidth of 0.11 ± 0.07 meV, indicating a low disordered crystal. The referee must note here that the non-zero intensity of the elastic line at $(0.25\ 0\ 4)$ is a tail of the nearby $(0\ 0\ 4)$ Bragg peak and not due to disorder or defects, as pointed out by the referee 2 and now rephrased in the main text. The lattice dynamics and the acoustic phonon dispersions along the $(h\ 0\ 4)$ and $(2+h\ 0\ 0)$ directions and their temperature dependence will be published in a separate manuscript.

IXS scan of the transverse acoustic phonon at 0.25 0 4 r.l.u.

5. In figure 3(a) and 3(c) authors have used three peak fitting whereas for 3(b) two peak fitting is used to demonstrate the absence of ω_1 at $h = 2.25$ at 110 K. But it seems that ω_1 with weak intensity is present in Figure 3(b). Authors can clarify on this.

We fully agree with the referee's suggestion and, indeed, there was a lot of discussion among all the co-authors about how to analyze the data at 110 K and $q=q_{\text{CDW}}$. We can give a value to the ω_1 , but the fitting is strongly biased and much affected by the large enhancement of the elastic line. As the referee points out, there might seem to be a weak intensity for ω_1 and in this particular case of $T= 110$ K and $q=q_{\text{CDW}}$ we would need to give ω_1 a finite value and force the fitting to converge to that value. However, we believe the phonon collapse we claim is well justified for the following reasons:

- DFT calculations nicely reproduce the temperature dependence of the ω_1 phonon including van der Waals forces, and those calculations predict a full collapse of the phonon at T_{CDW} .
- As shown in the inset of figure 3f, the phonon softening follows a mean field behavior, thus it is fair to expect that the ω_1 mode collapses to zero frequency at the CDW temperature.

We have modified the figure 3e and 3f (inset) and colored the 110 K point as cyan to highlight that the frequency of the phonon is not an experimental result but an extrapolated result.

6. As a general curiosity on the dispersion of ω_1 with h below CDW transition temperature, if the Kohn anomaly can be realized?

We have measured the phonon dispersion at 75 K, reproduced below (not included in the SI).

IXS scans at representative momentum transfers at 75 K. The large enhancement of the elastic line precludes the observation of the Kohn anomaly and the hardening of the soft mode.

As it can be seen, the large elastic line makes difficult to observe the hardening of the ω_1 phonon at $T < T_{\text{CDW}}$, as reported by F. Weber et al. in Phys. Rev. Lett. 107, 107403 (2011) for NbSe₂. Perhaps, selecting a different position in momentum space, where the elastic signal of the CDW is smaller can give a better hint about the referee's question.

Some minor comments:

7. Higher font size for text in Figure 1 (a) should be used for better visibility.
8. Some typo errors have been seen in the manuscript, which can be rectified during revisions.

We have corrected the font size of figure 1(a) and some typos.

Reviewer #2

The manuscript "Phonon collapse ..." by Diego et al presents an in-depth analysis of the CDW formation mechanisms in VSe₂. High-resolution experimental IXS data are supported by first-principles calculations of band structure, phonon spectrum including anharmonicity effects, and electron-phonon interaction. The main novelty of this work is the attribution of the CDW formation to the collapse of an acoustic phonon mode whose momentum coincides with the Fermi surface nesting. Other non-trivial findings include the EPI matrix-element effects, the role of anharmonicity in stabilization of high-temperature phase of VSe₂, etc. This is a wonderful in-depth work.

Response:

We are grateful to read the referee's response. We acknowledge him/her for his/her enthusiastic and positive review of our '*wonderful in-depth work*'. We also thank him/her for bringing up new comments, which help us improve the communication of our results. In the following, we reply one by one to the referee's questions.

I should however bring up to the author's attention a few points which might be elaborated:

- In connection with Fig. 1(b), it might be helpful to discriminate the elastic line from the elastic central peak in the experimental IXS spectrum;

We appreciate the referee's comment. We have reformulated the part of the text discussing the elastic line and the quasi-elastic central peak. In addition, in the supplementary information we have included a new graph to differentiate the elastic line from the quasi-elastic central peak.

- On page 3, the first paragraph, the authors drop that the momentum-space spread of the phonon mode indicates a substantial localization of the phonon fluctuations. How can such broadening be separated from that caused, in view of the acoustic-branch dispersion, by a drop of the phonon lifetime due to the EPI outburst near the CDW momentum?

The peak in the phonon linewidth (Fig. 1 (f)) is not so sharp as the referee seems to imply. Indeed, looking carefully, the region in which the phonon linewidth is enhanced goes roughly from slightly above $h=0.2$ to somewhat less than $h=0.3$. This nicely corresponds with the region in which the ω_1 peak is softened in Fig. 1 (e).

- The authors might be aware of the most recent ARPES work on NbSe₂ [F. Weber et al, Phys. Rev. B 97 (2018) 235122] which identifies clearly 3D character of the Fermi-surface pockets, denying any their nesting;

We thank the referee for bringing up this manuscript, which was out of our knowledge. The paper is now part of the discussion section.

- My main point of criticism is that the authors try to claim the phonon-mode collapse being the main mechanism of the CDW formation, whereas in reality it is highly entangled with the Fermi-surface nesting. Indeed, the momentum of this phonon itself is found, expectedly, exactly at the nesting vector determined by the electronic subsystem. Furthermore, the nesting by itself has a tendency to increase EPI. Finally, the authors claim that although the nesting-function peak does contribute to the phonon-energy renormalization, the peak of the EPI matrix element gives yet a larger effect; isn't this anomalous behavior of the latter exactly due to the electron-nesting feature of the electronic structure?

We understand referee's concern about this complex issue. First, we would like to stress here that the phonon collapse is the mechanism of the CDW formation. There is no doubt about that: the high-temperature phase becomes unstable thermodynamically exactly at the CDW temperature and distorts into the CDW phase undergoing a second-order phase transition. This is fully supported by our experimental and theoretical results.

However, this is unrelated to the origin of this softening. We guess the referee is actually asking about what the mechanism behind the softening is. As we argue in the manuscript, the softening is already apparent in the harmonic approximation. Somewhat thermal fluctuations and the consequent anharmonicity suppress this instability with increasing temperature and melt the CDW. Thus, the main question is to understand what is the origin of the harmonic softened phonon at q_{cdw} . As we argue in the manuscript both Fermi surface nesting and the momentum dependence of the electron-phonon matrix elements play a crucial role in the phonon self-energy (see equation for the self-energy in the supplementary information), which ultimately determines the harmonic phonon frequencies as well as the linewidth. The question is which of these effects is more important. In order to shine light on that question, we compare the phonon frequencies with the real part of the non-interacting susceptibility, which basically coincides with the real part of the phonon self-energy equation assuming that the electron-phonon matrix elements are equal to one; and the linewidth with the so-called nesting function, which again shares the same equation with the linewidth if the electron-phonon matrix elements are assumed to be one. The dependence on momentum is much stronger for both the linewidth and the phonon frequency than for the nesting function and the real part of the susceptibility. These suggests that the electron-phonon matrix elements play a crucial role in enhancing the softening and the linewidth, excluding the possibility that Fermi surface nesting alone suffices to

explain the softening. Indeed, as shown in Fig 4d, the ratio between the linewidth and the nesting function shows a very large dependence on momentum. If the matrix elements were constant on momentum, this ratio would be a constant even in the presence of nesting. As the ratio is strongly momentum dependent we can conclude that the electron-phonon matrix elements $(\langle n\mathbf{k} | \left[\frac{\partial V_{KS}}{\partial u_s^a(\mathbf{q})} \right]_0 | m\mathbf{k} + \mathbf{q} \rangle$, not affected by nesting) are strongly momentum dependent, and are enhanced in the region where the softening is observed. We thus believe that it is well-justified to say that the electron-phonon mechanism plays a major role in the phonon softening and, consequently, in the CDW formation.

In order to answer more explicitly to the questions raised by the referee, we may say that nesting is important but the magnitude of the momentum dependence of the electron-phonon matrix elements is the main source of the phonon softening around q_{cdw} .

In order to clarify these points further, the section of the manuscript "Role of electron-phonon interaction" has been extended and rewritten accordingly. For instance, we now explicitly mention the fact that the nesting function is basically the electron-phonon linewidth if the electron-phonon matrix elements are constant. Thus, the comparison between these magnitudes and the reason why we are plotting the ration between them in Fig. 4d, it is much clearer to the reader.

While the above points can certainly be elaborated, I am a bit hesitant regarding the publication recommendation. Apart from being too radical on the phonon-collapse role, the work is definitely solid and the results are reliable. However, it is narrowly focused on a particular material VSe₂ without a justified perspective to other cases of general interest. Furthermore, the manuscript is written for specialists and would hardly be accessible for a general reader. I am afraid that in Nature Comm, in the immense of popular biology or genetics papers, this great work would not score as much impact as in a more specialized physics journal such as Comm Phys or Phys Rev Lett.

We thank again the referee again for his/her positive suggestions to improve our manuscript. To give the work a broader scope, we have stressed the importance of a comprehensive understanding of the CDW formation in TMDs and monolayer materials. The referee may note that VSe₂ is being highly discussed recently in relation with the possible emergence of magnetism and the enhancement of the CDW at the atomic limit. Nature Communications has a tradition to publish papers related with ordering phenomena and charge density waves for specific compounds. See for instance:

- Chen, P., Chan, Y., Fang, X. *et al.* Charge density wave transition in single-layer titanium diselenide. *Nat Commun* **6**, 8943 (2015). <https://doi.org/10.1038/ncomms9943>.
- Lian, C., Zhang, S., Hu, S. *et al.* Ultrafast charge ordering by self-amplified exciton-phonon dynamics in TiSe₂. *Nat Commun* **11**, 43 (2020). <https://doi.org/10.1038/s41467-019-13672-7>.
- Chen, P., Pai, W., Chan, Y. *et al.* Emergence of charge density waves and a pseudogap in single-layer TiTe₂. *Nat Commun* **8**, 516 (2017). <https://doi.org/10.1038/s41467-017-00641-1>.
- Hellmann, S., Rohwer, T., Kalläne, M. *et al.* Time-domain classification of charge-density-wave insulators. *Nat Commun* **3**, 1069 (2012). <https://doi.org/10.1038/ncomms2078>.
- Flicker, F., van Wezel, J. Charge order from orbital-dependent coupling evidenced by NbSe₂. *Nat Commun* **6**, 7034 (2015). <https://doi.org/10.1038/ncomms8034>.

We, therefore, believe that our manuscript perfectly fits in a high-impact journal as Nature Communications.

Reviewer #3

The authors report inelastic x-ray scattering measurements of the phonons responsible for the charge density wave (CDW) transition in 1T-VSe₂, along with first-principles calculations to probe the origin of this transition. By studying momentum dependence of the inelastic spectra at different temperatures, one of the two observed acoustic modes is found to soften around the CDW wave vector quickly as the temperature decreases toward T_{CDW} , accompanied by an enhanced linewidth. From the large momentum space spread of the softening and further calculations showing strongly peaked phonon linewidth and electron-phonon matrix elements at the CDW wave vector, the authors conclude that the CDW instability is mainly driven by the electron-phonon interaction as opposed to pure Fermi surface nesting. Moreover, anharmonic phonon calculations suggest that the weak van der Waals forces are indispensable for describing the stabilization of the high-temperature structure or the melting of 3D CDWs in VSe₂. The present work is of interest for understanding the CDW origin in 1T-VSe₂, but there still remain some questions that need to be addressed, as described below.

We thank the referee for his/her critical reading of the manuscript and his/her positive comments. Below, we address the questions raised.

(1) On Page 1, the authors refer to the suppression of CDW with temperature in anharmonic phonon calculations as van der Waals melting in the title, based on the calculated result that the TCDW onset can only be reproduced when the van der Waals interactions are treated in the force calculations needed for the anharmonic SSCHA method. However, because the forces in supercells are calculated at $T = 0$ K, the real driving force for the melting of CDWs should still be the phonon-phonon interactions. On the other hand, in DFT-PBE calculations as adopted in this work, the van der Waals correction is essential for studying the properties of layered TMD materials and thus just a necessary ingredient. Furthermore, it is also confusing to attribute the role of van der Waals forces to an out-of-plane nature of the 3D CDW in VSe₂, which is argued to modulate the interlayer distance differently compared to 2D CDW systems like 2H-NbSe₂. To reinforce the so-called van der Waals melting, the current work mentions on Page 5 that anharmonic calculations explain the enhancement of T_{CDW} in monolayer VSe₂ but predict similar T_{CDW} values in bulk and monolayer NbSe₂ (Ref. 27, i.e. arXiv:2004.08147). Since anharmonic data on monolayer VSe₂ are not reported and whether or not the van der Waals correction was already considered in the NbSe₂ work remains unknown, it is hard to understand at present the difference in out-of-plane coupling between 3D and 2D CDW systems and its effect on the CDW transition.

The referee here is raising many interesting points. Let us answer point by point.

We completely agree with the referee that the CDW melts because of phonon-phonon interactions, or in other words by ionic entropy. These were clearly shown recently for NbSe₂ in Ref. 27, now published in PRL 125, 106101 (2020). However, as we pointed out in the reply to Reviewer #1, even if anharmonicity melts the CDW, the melting (below room temperature) can only be reproduced if van der Waals forces are included in the force calculations needed to run the SSCHA minimizations. Thus, van der Waals forces have a clear role in the melting of the CDW. Anyway, as a similar criticism was raised by Reviewer #1, we have updated the title of the manuscript.

Thus far, all the T_{cdw} predictions performed with the SSCHA method in TMDs have produced rather good agreement with experiments (see Refs. 25-27) without considering van der Waals (vdW) forces. Thus, it seemed that the role of van der Waals forces in the anharmonic renormalization of the phonons was minor, though we cannot discard that its inclusion could have improved the agreement with the experimental T_{cdw} . On the contrary, the SSCHA

approach without van der Waals forces completely breaks down in this system, as it cannot explain the melting of the CDW. We thus believe there is something special about the role of vdW forces in VSe₂, not so evident in other TMDs studied so far theoretically.

We attributed this special role of vdW to the 3D character of the CDW, i.e., to the fact that the CDW phase modulated the interlayer distance. This is not the case in NbSe₂. However, the referee is correct when noting that we do not have enough results to categorically make this statement. Therefore, we have smoothed a lot the discussion about the impact of the vdW on materials with a 3D character on the resubmitted manuscript.

(2) In the calculation details section of the supplementary information, it is noted that a mixed potential is used in DFT, phonon spectrum and electron-phonon calculations, i.e., a ultrasoft pseudopotential for V and a norm-conserving pseudopotential for Se. In consideration of the transferability of the potential and the fact that most theoretical works use pseudopotentials constructed for all atoms in a system within the same framework, the reasons why the authors adopt such a mixed potential need to be explained.

We understand the concern of the referee about the use of mixed pseudopotentials. However, a combination of ultrasoft and norm-conserving pseudopotentials was also used for the calculations in NbSe₂ (PRB 92, 140303(R) (2015) and PRL 125, 106101 (2020)), without any problem and showing good comparison with experiments. The referee should note that norm conserving pseudopotentials are a special case of ultrasoft pseudopotential. In this sense, any norm conserving is an ultrasoft in which the norm is conserved. Thus, there is absolutely no inconsistency in mixing ultrasoft and norm conserving potentials.

We have added a comment in the Supplementary Material to address this point.

REVIEWERS' COMMENTS

Reviewer #1 (Remarks to the Author):

The authors have performed the transport and magnetic measurements to show the temperature of the CDW transition. They have answered doubts raised and addressed the comments. A suggestion that authors may like to discuss the commensurate and incommensurate CDW in 1T-VSe₂. There are still some typo and technical writing error which can be taken care in proof reading.

For instance,

1. EDX is defined as "energy dispersive.." in text in SI and "electron dispersive.." in the caption of figure S1.
2. Line 15 of SI: its 'through', not 'thorough'.

Reviewer #2 (Remarks to the Author):

The revised version of the ms by Diego et al addresses essentially all my concerns, and I can recommend it for publication in Nature Comm. The only point where I feel the authors misunderstood my remark is about the k-space spread of the softening phonon mode, p. 3. Can this broadening been driven or at least contain a significant contribution from a reduction of the phonon lifetime caused by a sharp increase of EPI near the CDW momentum? I view this remark as optional that the authors might clear up in the final version of this interesting work.

Reviewer #3 (Remarks to the Author):

I appreciate the changes made in response to my previous report, which have eliminated my concerns about the role of van der Waals interactions and the use of a mixed potential. The manuscript is definitely improved, and the interesting results in the current work of VSe₂ will largely promote the research of CDW physics in other TMD materials. Therefore, I recommend its publication in Nat. Commun.

Response to referee's comments.

Referee 2 has raised the following question:

Can this broadening been driven or at least contain a significant contribution from a reduction of the phonon lifetime caused by a sharp increase of EPI near the CDW momentum?

Response

“Yes, as argued in the section ‘Role of electron-phonon interaction’, we believe that the large increase of the EPI is responsible for the phonon softening. Therefore, the EPI is also responsible for the momentum spread of the softening. In order to clarify this point further, we have slightly modified the last sentence of this part as:

Nevertheless, the q -range over which the phonon softens, $\Delta\text{q} \approx 0.075$ r.l.u., even if it coincides with increase of the electron-phonon linewidth, it is a factor of 3 less than in TiSe_2 [Web11Ti], where EPI and excitonic correlations are responsible for the structural instability and the CDW order, pointing to an intricate relationship between EPI and Fermi surface nesting scenarios in VSe_2 .

The manuscript has been updated accordingly and highlighted in red.